# OpenReview forum: "Capacity of Group-invariant Linear Readouts from Equivariant Representations: How Many Objects can be Linearly Classified Under All Possible Views?"
_NeurIPS.cc/2022/Workshop/NeurReps — NeurReps 2022 Oral_

### Official Review · Reviewer_oMSX · 2022-10-15
**Two new theorems related to linear classification**

**Confidence:** 3
**Soundness:** 3
**Presentation:** 4
**Contribution:** 3
**Overall Rating:** 7

**Summary:**

The authors present two theoretical results (with clean and concise proofs in the appendices) on an abstraction of linear classification tasks.

The core of the proofs is related to the separability of manifolds.
- They start with a lemma proving that some types of manifolds are linearly separable \\( \iff\\) the dataset of the corresponding centroids is linearly separable.
- Then they extend Cover's function counting theorem (Cover 1965) on the separability of linear sets. The finding is expressivity scales with the dimension of the subspace fixed by the group action.

They use their results to show how they can be applied to neural networks by showing that 2x2 max pooling operations reduce the network capacity, while average pooling does not.

**Questions:**

- In Theorem 2, the authors mention that the \\( \mathbf{r}^{\mu} \\) has to be drawn from a full rank Gaussian distribution to satisfy the condition. Do the authors have a sense of what would happen for other types of distributions?
- Continuing on question 1, are there other types of distributions for which the theorem still holds?

**Limitations:**

- The results on the pooling are interesting. While not a limitation per se it would be good to see this method used for more operations that can be found in modern neural networks.

**Recommended Decision:**

3: Accept

**Relevance:**

3: Solid fit

**Strengths And Weaknesses:**

# Strengths

- Clear problem formulation
- Clear theorems, demonstrated in details in the appendices
- Authors show that different types of pooling (max and average) respectively reduce and keep network capacity.
- Clear glossary in the appendices
- Great paper structure. Starting from background to theoretical results, to an impactful result that can be used by the practitioner (max pooling vs avg pooling)

# Weaknesses

It feels to me that there is a lot of content for an abstract when taking into account the appendices. It would have been easier to grasp all the contents in a proceedings track, and hinting at some of the proofs in the paper itself instead of leaving it in the appendices. Generally there is a lot of excellent content and proofs in the appendices and it would have been great to see in the paper.

**Submission Track:**

Extended Abstract (4 Page)

---

### Official Review · Reviewer_Xz98 · 2022-10-15
**Classification of manifolds generated by a group action**

**Confidence:** 4
**Soundness:** 4
**Presentation:** 3
**Contribution:** 2
**Overall Rating:** 6

**Summary:**

Classification capacity, or the number of objects which can be linearly classified correctly under random labels, is a proxy for the quality of the representations. The paper calculates the capacity for the special representations which are equivariant under group action, such as those which arise in convolutional layers of DNNs and group convolution and pooling layers of group DNNs. In a nutshell, capacity depends on $N_0$, the dimension of the fixed-point subspace of the group action’s linear representation. The theoretical predictions are demonstrated for the three relevant layer types.

**Questions:**

I don’t fully understand what assumptions are made on $\pi$, defined as an arbitrary linear representation of a compact group G.

 * Can you clarify under what conditions such a representation would exist? The assumption that such representation exist seems like the main weakness of the paper as it is true only for very specific layers (e.g., only convolutional layers with cyclic boundary conditions, group DNNs layers).
 * Can you clarify under what conditions such a representation would be linear? That is, is the fact that it’s *a linear* representation a fact rooted in some theorem, or a lucky case exploited here?


**Limitations:**

The authors discuss some non-trivial mathematical arguments, and one gets the feeling that some non-trivial limitations may still hide somewhere in the appendix details. My question above highlights one such case, so I’m not sure the authors have adequately addressed all the limitations of their work. An open discussion of the cases where such theory may apply may foster the development of specific architectures where the insight from current work applies or approximated methods which may address cases beyond the scope of the current method.

**Recommended Decision:**

3: Accept

**Relevance:**

4: Highly relevant

**Strengths And Weaknesses:**

Strengths
 * For representations which are equivariant under group action, where the action representation is linear, the paper makes a very clear statement.
 * It is a surprising finding that the dimension defined by the manifold’s centroid is enough to characterize the dimension of the manifold. This seems to stem from the “Invariance of the Haar Measure” and thus provides a unique property of such manifolds.
 * This is a great demonstration of how bringing ideas and methods from mathematics into neuroscience may trigger original contributions, which could not have grown “natively” inside the field.
 * The submission is of excellent quality; it is technically sound, and all claims are very well supported. But more importantly, it is clearly written despite the complicated synthesis of ideas from different fields.

Weaknesses:
 * It is not easy to find a representation of the kind discussed by the paper “in the wild”, as the assumptions are usually violated (e.g., the authors needed to modify DNN convolutions to have cyclic boundary conditions so that theory would make quantitive rather than qualitative predictions). Thus the significance of the results, in terms of importance and community interest, is limited.
 * As the conclusions of the papers are also limited to layers that exactly satisfy those assumptions, some of them provide the wrong intuition for other cases outside the scope of the paper. One such case is the effect of max-pooling which lower capacity here but have a more complicated effect on DNNs (see [Cohen, Chung, Lee, Sompolinsky 2020]).
 * Despite the authors' remarkable effort for clarity in their writing, there are few cases where math jargon is used without proper definition, providing no intuition and contributing nothing to the reader. In the main text, "trivial irrep” is an example of this, but additional examples are to be found in the appendix.

**Submission Track:**

Extended Abstract (4 Page)

---

### Official Review · Reviewer_CGNt · 2022-10-16
**A sound and important analysis of the theoretical separability of manifolds obtained by group transformations of inputs.**

**Confidence:** 4
**Soundness:** 4
**Presentation:** 4
**Contribution:** 4
**Overall Rating:** 9

**Summary:**

This paper explains how to predict the linear separability of manifolds created by group actions, based on the group representation qualities, extending the Cover theorem of separability of points to group-manifolds instead of points. It is shown that only the fixed point subspace of the group representation contributes to capacity (i.e. fraction of linearly separable dichotomies on a dataset of P group-manifolds). The predictions of the theory are matched by simulations on equivariant neural networks.

**Questions:**

What are practical implications of these findings for the design of (equivariant) deep networks?


**Limitations:**

Limitations are adequately addressed.

**Recommended Decision:**

3: Accept

**Relevance:**

4: Highly relevant

**Strengths And Weaknesses:**

Significance: this work is important to understand how the property of equivariance in neural networks affects their representational capacity.

Originality: this extended abstract is based on a recently published work, but apart from this recently published work I have not seen any similar contribution elsewhere.

Quality: Although I have not checked in detail the mathematical proofs, they look sound to me and and the intuitions behind lemma 1 and theorem 2  are nicely explained geometrically in the appendix.

Clarity: Claims are very clear and well explained.

**Submission Track:**

Extended Abstract (4 Page)

---

### Decision · Program_Chairs · 2022-10-21

Accept (Oral)